# In Search of Concrete Outcomes—A Systematic Review on the Effectiveness of Educational Interventions on Reducing Acute Occupational Injuries

**DOI:** 10.3390/ijerph17186874

**Published:** 2020-09-20

**Authors:** Jim Li, Max Pang, Jennifer Smith, Colleen Pawliuk, Ian Pike

**Affiliations:** 1MD Undergraduate Program, Faculty of Medicine, University of British Columbia, Vancouver, BC V6T 1Z3, Canada; mpang@alumni.ubc.ca; 2BC Injury Research and Prevention Unit, Faculty of Medicine, University of British Columbia, BC Children’s Hospital, Vancouver, BC V6H 3V4, Canada; jsmith@bcchr.ca (J.S.); ipike@bcchr.ca (I.P.); 3BC Children’s Hospital Research Institute, Faculty of Medicine, University of British Columbia, BC Children’s Hospital, Vancouver, BC V6H 3V4, Canada; cpawliuk@bcchr.ca; 4Department of Pediatrics, Faculty of Medicine, University of British Columbia, Vancouver, BC V6H 3V4, Canada

**Keywords:** accident prevention, educational intervention, injury prevention, occupational injuries, preventive medicine, public health, safety management, safety training, systematic review

## Abstract

Education is a common strategy used to prevent occupational injuries. However, its effectiveness is often measured using surrogate measures instead of true injury outcomes. To evaluate the effectiveness of workplace educational interventions, we selectively analyzed studies that reported injury outcomes (PROSPERO ID: CRD42019140631). We searched databases for peer-reviewed journal articles and sources of grey literature such as abstracts, registered trials, and theses published between 2000 and 2019. Studies on educational interventions that reported fatal or non-fatal occupational injury outcomes were selected. Two reviewers independently and in duplicate screened the studies, extracted data, and assessed risk of bias. Heterogeneity in the data precluded meta-analysis, and the results were reviewed narratively. In total, 35 studies were included. Of which, 17 found a significant reduction in injuries, most of which featured a multifaceted approach or non-didactic education. The remaining studies either described equivocal results or did not report statistical significance. Overall, interventions in the manufacturing industry were more effective than those in the construction sector. Risk of bias among included studies was moderate to high. In conclusion, educational interventions could be an effective part of multifaceted injury prevention programs. However, over-reliance on didactic education alone is not advised.

## 1. Introduction

Occupational injuries are a major threat to the health and well-being of workers. Each year, an estimated 380,000 workers lose their lives due to workplace injuries globally, and over 370,000,000 suffer non-fatal injuries [1]. In 2015, the International Labour Organization estimated that a total of USD $2.8 trillion is lost each year due to the direct and indirect effects of occupational hazards globally, accounting for 4% of the world’s GDP [2,3].

Many prevention strategies have been implemented in recent decades in an effort to stem occupational injury levels. These initiatives can be broadly categorized into the “Three E’s of Injury Prevention”, education, enforcement, and engineering [4]. Education refers to providing information and training to elicit behavioral change, enforcement includes laws and regulations to deter unsafe practices, and engineering entails modifying the environment and/or the task to enhance safety. Since educational interventions theoretically work by changing employee attitudes, behavior, and knowledge, their effectiveness is often assessed based on such surrogate measures, instead of direct injury outcomes [5]. This assumption has been criticized for being overly simplistic, as surrogate measures are not necessarily representative of true injury rates, thereby making the data difficult to interpret with certainty [6,7]. Various reviews on educational interventions have been challenged to make concrete recommendations due to the lack of studies that report injury outcomes [6,7,8,9]. This highlights a gap in the literature, as the true effectiveness of education on reducing workplace injuries remains unclear.

Although some previous systematic reviews have been selective, by only including studies with injury outcomes, they are limited to specific industries, such as construction or agriculture [10,11], which precludes generalization of the results to occupational injuries as a whole. Therefore, we endeavored to systematically review studies across all industries describing workplace educational interventions aimed at preventing occupational injuries, which also reported worker safety outcomes.

## 2. Materials and Methods

The review protocol is registered in PROSPERO (ID: CRD42019140631). The review was conducted in accordance with the preferred reporting items for systematic reviews and meta-analyses (PRISMA) guideline [12].

### 2.1. Types of Studies

Randomized controlled trials (RCTs), non-randomized controlled studies (NRSs), and uncontrolled before-and-after trials (uCBAs) were eligible for inclusion in this review. We expected a limited number of RCTs due to various logistical and ethical barriers inherent within the context of workplace health and safety education initiatives. For instance, educational interventions using individual-level randomization in the workplace may be prone to contamination between groups, since workers can freely communicate amongst each other. Additionally, workplace-based cluster randomization faces challenges associated with recruiting a sufficiently large sample to be robust against confounding variables. Therefore, we decided to include a number of non-randomized study designs, such as controlled before-and-after studies (CBA), interrupted time studies (ITS), historically controlled studies, prospective cohort studies, and uncontrolled before-and-after studies (uCBAs).

We defined a CBA study as a controlled trial in which the experimental and control groups were not assigned in a random fashion. A historically controlled study is similar to a CBA study, with the difference that the control was a group of participants in the past. In the absence of a control group, if a study measured three time points before and after the intervention to analyze for temporal trends, then it qualified as an ITS study. A prospective cohort study was defined as a study in which the investigators enrolled participants after exposure to a factor associated with the outcome, but before any outcome(s) of interest had developed. Finally, an uCBA study implements an intervention without including a control group [13]. 

Our decision to include uncontrolled before-and-after studies was due to three reasons. First, in the occupational context, controlled trials may be difficult to perform. Second, since the literature connecting educational interventions to injury outcomes is already limited to begin with, over-restriction of our search criteria may have limited our ability to draw any conclusions at all. Third, we anticipated that a sizeable proportion of the literature on this topic may be the result of quality improvement projects which were later published. By including these uncontrolled trials, we could gain additional insight into educational interventions used in a practical context.

### 2.2. Types of Participants

This review included studies of paid adult (≥18 years of age) employees from all industries. Temporary workers or workers employed outside of a legal contract were also included.

### 2.3. Types of Interventions

Educational interventions implemented with the intention of preventing occupational injuries were included. Multifaceted interventions with a prominent educational component were included as well. We included didactic education in addition to non-traditional methods, such as interactive discussions, peer feedback, hazard recognition, and social marketing/awareness campaigns.

### 2.4. Types of Outcome Measures

We included studies that measured the change in fatal or nonfatal injury rate following educational intervention(s). As an operational definition, we used the following modified definition from The Injury Chartbook by the World Health Organization (WHO) [14,15], “occupational injury is a body lesion at the organic level, resulting from acute exposure to energy in the work environment (mechanical, thermal, electrical, chemical or radiant) in amounts that exceed the threshold of physiological tolerance. In some cases (e.g., drowning, strangulation, freezing), the injury results from an insufficiency of a vital element.” Since we anticipated that operational definitions for injury would vary by study, we considered all reasonably congruent definitions. All collection methods of quantifiable injury data were eligible for inclusion. These included, but were not limited to, self-report, company databases, and workers’ compensation records.

### 2.5. Exclusion Criteria

We excluded studies published before 2000 to ensure the recency of our results. The year 2000 was selected due to the rapid automation of industries which caused substantial changes in the proportion of white-collar, operative/laborer/service, and agricultural workers that did not decelerate until 2000 [16,17]. This coincided with a shifting paradigm for occupational safety education, which better reflected the needs of the changing workplace [18,19]. We also excluded non-English studies due to limited translation resources. Studies that measured injury rates outside of the occupational context were excluded. For our study, “occupational context” is defined as both the workplace and commute to and from work. Further, we excluded occupational diseases, infections, intentional injuries, and musculoskeletal disorders due to chronic exposure. 

### 2.6. Information Sources

We searched the following electronic databases up to May 2019: Embase, Ovid MEDLINE, NIOSHTIC, Web of Science Core Collection, and the Cochrane Central Register of Controlled Trials (CENTRAL). In order to avoid publication bias favoring positive results, grey literature such as theses, abstracts, and registered clinical trials were also included in our study [20]. We searched Google Scholar, Google, WorldCat, ProQuest, NDLTD (Networked Digital Library of Theses and Dissertations), and ClinicalTrials.gov for grey literature. We also manually searched the reference lists of all included studies to locate additional papers. All searches were conducted with the guidance of a biomedical librarian. The general PICO (population, intervention, context, and outcome) search structure for the databases is shown in Table 1; the detailed search strategy is listed in Appendix A.

### 2.7. Selection of Studies

The titles and abstracts of potentially relevant studies were screened by two review authors independently and in duplicate. Disagreements about inclusion occurred in less than 5% of all records screened, and were resolved by discussion until consensus was reached. In the case of a disagreement that could not be resolved by discussion, a third author made the final decision. Full text articles of the shortlisted studies were then reviewed independently and in duplicate by two authors against the inclusion and exclusion criteria. The same process for resolving disagreements was applied, and reasons for exclusion were documented. Reference lists of included papers and relevant systematic reviews were screened to identify additional studies. The titles of relevant grey literature were searched on Google Scholar to potentially locate their respective full papers.

### 2.8. Data Extraction and Management

Two review authors independently extracted data from all included articles. We designed a data extraction form in order to standardize the process between the two reviewers. The following data were collected: study design, setting (nationality and industry, as defined by the North American Industry Classification System [21]), participants (sample size, occupation, and distribution between experimental and control groups), description of the intervention(s), timeline of intervention, outcome (rate ratio, difference in means, etc.), sources of funding, and conflicts of interest.

### 2.9. Assessment of Risk of Bias in Included Studies

Two review authors assessed risk of bias in all included studies independently and in duplicate. The quality of RCTs was assessed using the revised Cochrane risk-of-bias tool for randomized trials (RoB 2) [22]. Quality of non-randomized controlled studies was assessed using the Cochrane risk of bias in non-randomized studies of interventions (ROBINS-I) [23]. Disagreements arose in about 10% of assessments but were all resolved by discussion without resorting to a third reviewer. The risk of bias for conference abstracts could not be assessed due to limited information, and thereby they were categorized as “uncertain risk of bias.” Uncontrolled before-and-after trials were all categorized as “high risk of bias” since they are especially prone to confounding factors and regression to the mean [24].

### 2.10. Data Synthesis

Meta-analysis was not attempted due to heterogeneity in the target populations and study designs, as well as an insufficient number of studies from most industries. Since each industry has its unique profile of injury mechanisms and risk factors, we expected the nature of educational interventions to vary accordingly [25]. Therefore, qualitative assessment was performed at an industry-level, by considering the number and quality of studies, effect significance, and type of education involved. Included studies with a “critical” risk of bias were not factored into any narrative synthesis, as per Cochrane guidelines [23].

## 3. Results

### 3.1. Results of the Search

The search yielded 4492 records, of which 3973 were from electronic databases and 519 were identified through sources of grey literature. (Later manual checking of reference lists from relevant systematic reviews and included studies yielded an additional 5 records.) After removing the duplicates, 3613 remained. After screening their titles and abstracts, 48 studies were shortlisted for full-text review. Of these 48 papers, 38 met inclusion criteria (Figure 1), and 10 were excluded. The reasons for exclusions are listed in Appendix B. The 38 included papers represented 35 unique studies. In cases where multiple papers described the same study, only the most comprehensive publication was retained for further analysis. The redundant papers are also listed in Appendix B.

### 3.2. Study Characteristics

After full-text review, 35 studies were included [Figure 1]. This included eight RCTs [26,27,28,29,30,31,32,33], six CBA studies [34,35,36,37,38,39], four ITS studies [40,41,42,43], one historically controlled study [44], two prospective cohort studies [45,46], and fourteen uCBA studies [47,48,49,50,51,52,53,54,55,56,57,58,59,60]. Studies describing multiple similar interventions were considered as one study [48,49,60]. Of the 35 studies, 14 were from the USA [28,29,33,34,38,39,44,46,47,50,51,53,59,60], 5 from Italy [37,41,42,43,48], 2 from Australia [31,57], 2 from Denmark [26,52], and 1 each from Belgium [27], China [30], India [32], Cuba [35], Mexico [36], Germany [40], France [45], Nigeria [58], Finland [49], Egypt [54], Zimbabwe [55], and Canada [56].

All included studies were published between 2000–2018. Unless otherwise specified, all studies were published as a full paper. Characteristics of all included studies are summarized in Table 2.

There were 4 RCTs, 4 CBAs, 2 ITSs, 1 historically controlled study, 1 prospective cohort, and 5 uCBAs which demonstrated a significant reduction of injuries [30,31,32,33,35,36,37,38,40,42,44,45,52,54,55,57,60]. However, 4 RCTs, 2 CBAs, 2 ITSs, 1 prospective cohort, and 9 uCBAs produced equivocal or non-significant results [26,27,28,29,34,39,41,43,46,47,48,49,50,51,53,56,58,59]. Sources of funding and conflicts of interest for included studies are shown in Appendix C. 

### 3.3. Risk of Bias in Included Studies

Of the eight RCT studies, five were judged to be at moderate risk of bias using the RoB 2 tool, and three were at high risk of bias (Figure 2).

Using the ROBINS-I tool, of the five CBA studies that were published as a full paper, four were at serious risk of bias and one was at critical risk of bias. Of the three ITS studies that were published as full papers, two were at moderate risk of bias, and one was at critical risk of bias. The historically controlled study and both of the prospective cohort studies were at serious risk of bias. The two studies with critical risk of bias will not be further discussed in the narrative synthesis [36,41] (Figure 3).

As discussed previously, all 14 uCBAs were automatically assigned a high risk of bias [24]. Risk of bias assessment was not attempted for the remaining abstract (ITS) and the grant report (CBA) due to insufficient information.

## 4. Discussion

Despite widespread use of education in occupational injury prevention programs, limited reviews exist on assessing injury outcomes in recent years. Hence, we aimed to narratively summarize the effects of educational interventions on occupational injury outcomes through this study. We systematically searched multiple databases and various sources of grey literature. A strength of our study is that we included grey literature, including abstracts, theses, and grant reports to avoid publication bias. However, a necessary trade-off was that their interventions and results were not always described in detail. Further, by considering a broad range of injuries across all industries, the results of this study could be more generalizable. To the best of our understanding, this is the first systematic review of its kind that is not limited to a particular industry or type of injury. 

Our results reveal modest evidence that educational interventions have a protective effect against occupational injuries. However, the overall risk of bias was moderate to high among all included studies. In addition, the majority of all studies were performed in two industries, construction and manufacturing. Multifactorial and non-didactic educational interventions were generally more effective than didactic education. This is especially evident in the manufacturing sector, which employed the former strategies more frequently. 

### 4.1. Agriculture, Forestry, Fishing, and Hunting

The agricultural sector experiences the highest rates of non-fatal injuries among all US industries [61]. However, neither of the RCTs included were able to demonstrate a significant reduction in injuries, despite utilizing multiple forms of education, such as didactic and interactive teaching, plus behavioral-based incentives [26,29]. Both studies were conducted in developed countries with a relatively high degree of legislative and engineering support. While this gives us some insight into the effect of education in developed countries, it limits our ability to extrapolate the findings to developing countries, where most of the world’s farming population resides [62]. Our findings corroborate a meta-analysis by Rautiainen et al., who concluded that there was no evidence suggesting benefit from the use of educational interventions alone in the agricultural context, and that more high-quality studies, such as RCTs or ITSs, should be conducted for behavioral interventions [11].

### 4.2. Arts, Entertainment, and Recreation

Only one RCT was identified, targeting sunburn reduction among swimming pool staff [33]. This limits our ability to generalize the results to the rest of this sector. Nonetheless, this study is notable for using a placebo intervention in the control group, in which participants learned about child injury prevention. This strategy could help reduce the risk of bias associated with the impracticality of blinding participants in educational interventions.

### 4.3. Construction

Although a large number of studies were identified in construction, not all of them were methodologically sound. Our search revealed one CBA [39], one ITS which was removed from narrative synthesis due to critical risk of bias [41], one prospective cohort [46], and five uCBAs [50,51,52,53,59]. The overall results were not encouraging. Only one uCBA study found a statistically significant effect following an educational intervention [52]. Thus, there may be challenges in influencing the construction sector through purely educational means. This echoes a previous meta-analysis by van der Molen et al., which showed no strong evidence to suggest that safety campaigns alone have a protective effect on construction workers [10]. There may be a potential synergistic effect between educational and regulatory interventions, which may serve as a potential direction for future research [63,64]. It should also be noted that construction studies besides Spangenberg et al. [52], Evanoff et al. [50], and Kidd et al. [39] featured interventions that were almost exclusively didactic in nature. Interestingly, Spangenberg et al. was also the only study to find a significant effect, and Evanoff et al. found a significant effect before adjusting for covariates. This may indicate that non-didactic education is more effective, hence suggesting a direction for future research.

### 4.4. Educational Services

One RCT was identified [27], which did not find a significant benefit from educational intervention. Since the study was conducted in only one region in Belgium, risk of cross-contamination may have skewed the results. Moreover, since its results were analyzed on a per-protocol basis, it limits our ability to extrapolate results to the real world, where non-adherence may be prevalent.

### 4.5. Health Care and Social Assistance

Only one uCBA abstract was identified [47], which did not state statistical significance. There was insufficient information for conclusions to be drawn for this sector of the economy.

### 4.6. Manufacturing

Manufacturing had the largest number of included studies: two RCTs [30,31], three CBAs [36,37,38] (one of which was removed from narrative synthesis due to critical risk of bias [36]), three ITS studies [40,42,43], and four uCBAs [48,54,55,56]. Of which, all of the RCT and CBA studies, one of the ITS studies [42], and two of the uCBA studies [54,55] showed significant injury reductions. Interestingly, one commonality among the majority of effective interventions in manufacturing was that they employed either multifactorial approaches or educational methods that were not purely didactic [30,31,38,42,54,55]. Despite the possibility of cross-contamination favoring the controls, these interventions still resulted in significantly positive results, which may indicate that such efforts are especially effective. Another promising aspect is that these studies represent a range of developed and developing countries, thereby increasing the generalizability of the results.

### 4.7. Mining, Quarrying, and Oil and Gas Extraction

Three studies were included, one each of RCT, CBA, and uCBA. Overall the results are encouraging as the RCT and uCBA demonstrated significant protective effects [32,60], while the CBA [34] showed promising effects in the protective direction (significance not stated). The relatively large effect sizes reported in these studies give us cautious optimism that education is an effective component in future mining interventions.

### 4.8. Public Administration

An RCT and an historically controlled study were included [28,44]. Both of which saw significant benefits in some but not all outcomes measured [Table 2]. Despite considerable cross-contamination between groups in the RCT, the authors persisted with an intention-to-treat (ITT) design which made the results more robust against non-adherence [28]. However, research in this sector is still lacking.

### 4.9. Transportation and Warehousing

One CBA and one uCBA were included [35,57]. Both studies only focused on a small subset of the overall population (i.e., stevedores and truck drivers with sleep disorders), which severely limits the generalizability of any conclusions drawn. As one of the most common causes of preventable fatal injuries, additional investigations in this field is needed [61]. 

### 4.10. Utilities

Salminen found an increase in injury numbers after implementing an anticipatory driving intervention for electricians [49]. In addition, although traffic-related incidents were reduced following another group discussion intervention, a paradoxical increase in non-traffic injury rates occurred. These could be rationalized by considering that the study was uncontrolled with a short follow-up period, and that behavioral changes may take a long time to mature. If the trial was more robust, the principles of driving safety could be extrapolated to other industries which involve driving. Another uCBA in the electrical sector was an abstract with relatively little information regarding injury outcomes, so it was not possible to draw conclusions from it [58]. Overall, educational interventions in this industry are understudied and greatly limited in scope.

### 4.11. Observations across Industries

Overall, the methodological quality of the included studies was poor, as shown in Figure 2 and Figure 3. Even among the RCTs, none were at a low risk of bias. This observation can be rationalized by first considering the unique context of the occupational injury prevention scenario. Randomization in this setting may face ethical questions associated with denying potentially life-saving interventions from workers. In addition, even if that can be addressed, the study may run into logistical challenges when trying to recruit enough workplaces for randomization, as the prospect of being placed into the control group may discourage employers. On the other hand, if randomization was performed on an individual basis within individual workplaces, then serious concerns with cross-group contamination would be inevitable. Further, after randomization it would be practically and ethically unfeasible to prevent workplaces from implementing additional interventions, which would confound the results. This is especially pertinent with educational interventions, as changes in beliefs, attitudes, and work culture take time, during which confounding interventions may occur. Moreover, with an educational intervention, true blinding of the participants and instructors is practically challenging. For these reasons, sometimes the best available option is to implement a non-randomized trial or an RCT with considerable limitations. This has been echoed in previous Cochrane reviews [10,11]. As such, we have taken these constraints into account when making our recommendations. 

To facilitate future RCTs, we suggest the following strategies. Conducting studies proactively (i.e., before injury rates become alarmingly high) could encourage employers to accept the risk of having their workplace being assigned to a control group. As an added benefit, it is less likely that employers will implement confounding interventions of their own during the study. Alternatively, offering crossover study designs would remove the disincentive of potentially being assigned to a control, while at the same time addressing ethical concerns of withholding beneficial interventions from workers. Finally, control groups may receive standard training or placebo intervention on another topic to enhance the blinding process. Such strategies could bring about additional high-quality studies in the field, which could in turn allow for further conclusions to be drawn.

There were four studies which attempted to indirectly prevent injuries through addressing associated conditions such as obesity, skin cancer, and sleep disorders [28,31,33,57]. Of which, Geller et al., Morgan et al., and Sullivan et al. were RCTs with significant results in at least some of their outcome measures. This suggests that education does not necessarily have to be directly focused on injuries. The fact that these three RCTs were performed in diverse settings supports the generalizability of this notion. 

It should be noted that among the interventions which demonstrated a significant protective effect against occupational injuries, most featured either multifactorial strategies alongside education [33,42,44,54,57] or educational approaches that were not purely didactic in nature [30,31,32,38,52,55]. This suggests that creative and multifaceted designs should be utilized when designing future injury prevention programs, especially in the manufacturing industry, where many of the aforementioned studies took place. While it may seem obvious that a multifactorial approach would have greater success than education alone, there is a potential synergistic effect as every link along the chain of safety is strengthened [64]. On the other hand, the effects of implementing didactic education on its own are limited, as evidenced by studies in the construction sector, which may not justify its resource and opportunity costs. 

In general, developing countries were underrepresented amongst the included studies. Only six studies were done on workers in developing economies [30,32,35,54,55,58]. This is concerning, as the majority of occupational injuries occur in these settings [1]. Interestingly however, all of those studies showed either a significant benefit [30,32,35,54,55] or a protective effect, without stating statistical significance [58], which suggests that education may be more effective in low-resource settings. This could be rationalized by considering that legislative, administrative, and engineering interventions presumably already exist in developed countries, thus the effects of additional education would be dwarfed in comparison. Conversely, in developing countries, those measures may not be as robust, which leaves the potential for education to impart a more pronounced effect. 

### 4.12. Limitations

Due to limited resources, we were unable to include non-English language studies, which may have introduced a language bias. Additionally, since some educational interventions were implemented as part of a multifaceted program, it is sometimes difficult to discriminate what effects, if any, that the educational components truly imparted. However, this would not invalidate these studies as the purpose of our review is to address the pragmatic question of whether or not education is effective within the context of a real-life work environment, where there will inevitably be some degree of concurrent interventions. Due to heterogeneity in study designs and a limited number of RCTs, it was not possible for a meta-analysis to be performed and therefore quantitative conclusions cannot be drawn. Finally, since most studies were conducted in developed nations, a caveat is that our findings are not necessarily generalizable in the setting of developing economies.

## 5. Conclusions

Educational interventions are effective when implemented as part of a multifactorial approach or in a non-didactic fashion. This is especially true in the manufacturing sector. Caution should be advised when implementing didactic education on its own to prevent occupational injuries, especially in construction. Nevertheless, it could still be an effective component of a multifactorial approach. Additional high-quality studies in underrepresented industries and developing countries are needed to better understand the effectiveness of education in their respective settings. In the future, researchers could address barriers to RCTs by implementing interventions proactively, using crossover designs, and providing controls with standard training or placebo intervention.

## Figures and Tables

**Figure 1 ijerph-17-06874-f001:**
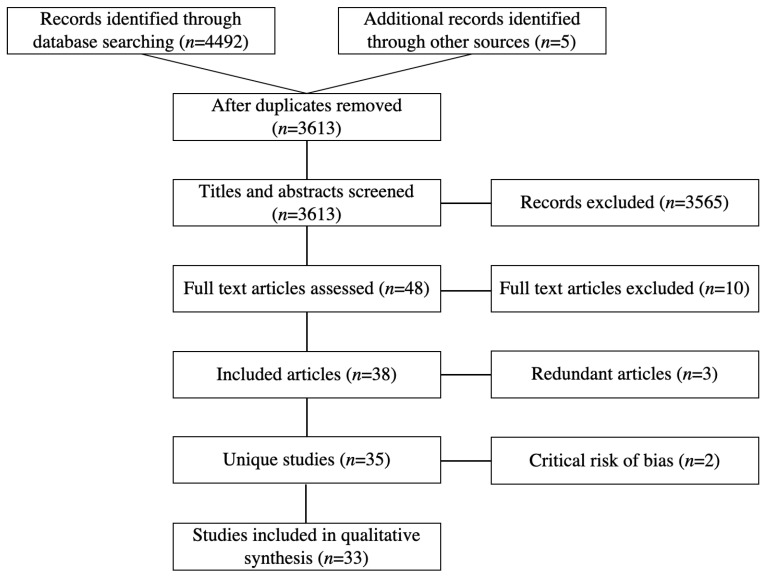
Modified preferred reporting items for systematic reviews and meta-analyses (PRISMA) flow diagram.

**Figure 2 ijerph-17-06874-f002:**
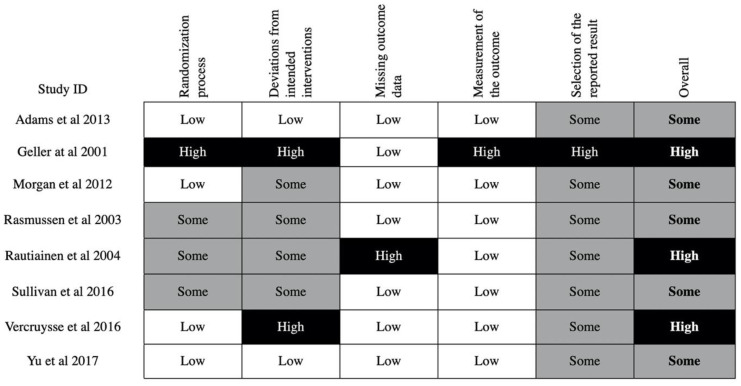
Risk of bias in randomized controlled trial (RCT) studies—risk of bias was calculated using the risk of bias (RoB) 2 tool. The level of bias for each of the 5 domains are shown above for the 8 RCT studies.

**Figure 3 ijerph-17-06874-f003:**
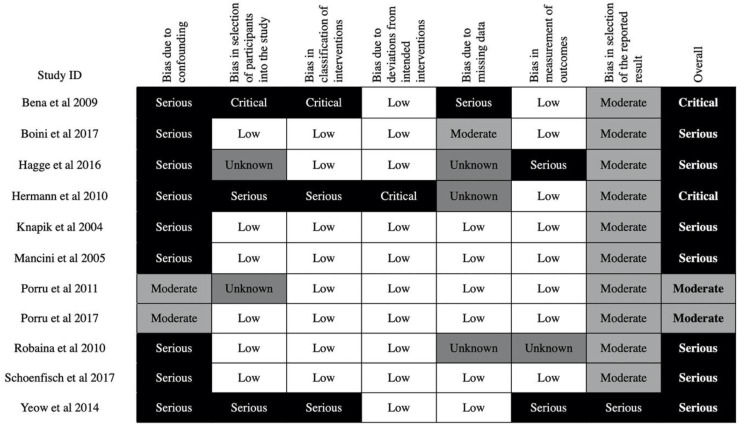
Risk of bias in non-randomized controlled studies—risk of bias was calculated using the ROBINS-I tool. The level of bias for each of the 7 domains are shown above for the 11 non-randomized controlled studies. Note that the terminology used between RoB 2 and ROBINS-I is slightly different.

**Table 1 ijerph-17-06874-t001:** General search topics included in the final PICO (population, intervention, context, and outcome) framework.

Population/Problem	Intervention	Context	Outcome
Injury	Education	Work	Injury prevention
Wounds	Program	Occupation	Injury reduction
Fatality	Social Marketing	Employee	Insurance claims
Disability	Primary Prevention	Industry	Absenteeism

**Table 2 ijerph-17-06874-t002:** Summary matrix of all included studies: tabulated data include information on industry, study design, participants, intervention, and outcomes.

Agriculture, forestry, fishing and hunting.	**Study ID**	**Type of Study**	**Participants**	**Timeframe**	**Intervention(s) and Control (If Applicable)**	**Outcomes**
	Rasmussen et al., 2003 [26]	RCT randomized at the farm level	Farmers in Denmark Intervention: 99 farms, *n* = 498 Control: 102 farms, *n* = 492	Duration: November 1995 to July 1997 Extended follow up: N/A	Safety audits with feedback, and a 1-day safety course featuring didactic and participatory education; controls received no safety course	No significant effect in preventing injuries compared to control, even when adjusting for seasonal variation—**30% reduction in injury rate (95% CI: 21% increase—60% reduction)**
	Rautiainen et al., 2004 [29]	Matched pairs cluster RCT, randomized at the farm level	Farmers in the USA Intervention: *n* = 152 farms Control: *n* = 164 farms	Duration: 1998 to 2003 Extended follow up: N/A	Annual clinical screening, access to didactic materials, and a monetary incentive for farmers who received a safety audit score; controls received no intervention	No significant effect in preventing injuries compared to control, even when adjusting for confounding factors—rate ratio of injury rates (intervention vs. control) was **1.08 (96% CI: 0.86–1.37)**
Arts, Entertainment, & Recreation	Geller et al., 2001 [33]	Matched-pairs cluster RCT, randomized at the pool level	Pool staff in the USA Intervention: *n* = 142 Control: *n* = 78	Duration: summer 1999 Extended follow up: N/A	Didactic education on sunburns and skin cancer, plus engineering and personal protective equipment support; the control group received placebo intervention on child injury prevention.	Significant effect in preventing sunburns compared to control—**1.42** was the average number of sunburns among intervention participants, and **2.07** among control participants (*p* < 0.05)
Construction	Bena et al., 2009 [41] ^1^	Interrupted time series	Construction workers in Italy Basic modules: *n* = 2320 or 2375 worker jobs Specific modules: *n* = 1158 or 1164 worker jobs	Duration: July 2002 to December 2005 Extended follow up: N/A	Didactic module-based education, basic modules for all workers and specific modules for specific jobs	No significant effect in preventing injuries—estimated **6% decrease** in the frequency of injuries per 10,000 days of exposure (*p* = 0.08)
	Kidd et al., 2007 (grant report) [39]	Controlled before and after	Construction workers in the USA Intervention: *n* = 260 Control: *n* = 95	Duration: 1995 to 1999 Extended follow up: N/A	Participatory training for fall prevention and back injury prevention	No significant differences in the number or cost of injury claims compared to control
	Schoenfisch et al., 2017 [46]	Prospective cohort	Carpenters in the USA Exposed: *n* = 377 with OSHA-10 training, *n* = 76 with OSHA-30 training Unexposed: *n* = 17,106	Duration: 2000 to 2008 Extended follow up: N/A	The Occupational Safety and Health Administration (OSHA) offered 10- and 30-h didactic injury prevention courses	No significant effect in preventing injuries—the rate ratio of injuries in trained vs. untrained workers was **0.87 (95% CI: 0.72–1.06)**
	Evanoff et al., 2016 [50]	Uncontrolled Before and After	Carpenters in the USA, *n* = 2291	Duration: 2007 to 2009 Extended follow up: N/A	Didactic and participatory education on fall prevention	No significant effect in preventing injuries, once covariates are accounted for—incidence rate of falls (95% CI) **pre-intervention: 18.3 (15.5–21.1) vs. post-intervention: 14 (11.7–16.2)**
	Darragh et al., 2004 [51]	Uncontrolled Before and After	Construction workers in the USA *n* = 97 companies	Duration: 1997 to 1998 Extended follow up: N/A	Three-hour safety training and orientation, plus access to optional 10-h OSHA training courses	No significant effect in preventing injuries—the risk ratio before and after intervention was **0.97 (95% CI: 0.5–2.0)**
	Johnson & Ruppe 2002 [59]	Uncontrolled before and after	Construction workers in the USA, *n* = 55	Duration: 1998 to 1999 Extended follow up: N/A	Didactic education, plus stress management and spirituality sessions	Although injury rates decreased, statistical significance was not stated
	Spangenberg et al., 2002 [52]	Uncontrolled before and after	Construction workers in Denmark, sample size not stated, but estimated 6.8 million person-hours worked	Duration: 1996 to 1998 Extended follow up: N/A	Didactic education, safety-based monetary incentives, and a social marketing campaign	Significant effect in preventing injuries, once concurrent changes in construction intensity are accounted for—**25% reduction in injuries (95%CI: 1–43%)**
	Williams et al., 2010 [53]	Uncontrolled before and after	Construction workers in the USA (all Latino males), *n* = 313	Duration: 1 day Extended follow up: 2–6 months	Didactic education on topics pertinent to Latino immigrant workers	Although injury rates decreased by **24.0%**, statistical significance was not stated
Education Services	Vercruysse et al., 2016 [27]	Cluster RCT, randomized at the school level	Physical education teachers in Belgium Intervention: *n* = 29 teachers from 19 schools Control: *n* = 26 teachers from 20 schools	Duration: September 2013 to June 2014 Extended follow up: N/A	Didactic and participatory education, access to didactic videos; controls did not receive any intervention	No significant effect in preventing injuries compared to control—**0.20 (95% CI: 0.06–0.61)** work injuries per 1000 h in the intervention group; **0.54 (0.24–1.18)** work injuries per 1000 h in the control group
Health Care & Social Assistance	Koblesky 2017 (abstract) [47]	Uncontrolled before and after	Blood center employees in the USA, 2010 to 2014 Sample size not stated	Duration: 2011 to 2014 Extended follow up: N/A	Didactic education and social marketing, plus administrative changes	Although the number of injuries, workers’ compensation claims, and days away from work decreased, statistical significance was not stated
Manufacturing	Yu et al., 2017 [30]	Paired cluster RCT	Factory workers in China Intervention: *n* = 966 from 30 experimental factories Control 1: *n* = 966 from the same 30 experimental factories Control 2: *n* = 1706 from 30 control factories	Duration: June 2008 to November 2009 Extended follow up: 12 months	Participatory education featuring a workplace inspection exercise followed by discussion on implementable actions; controls received didactic education only	Significant effect in preventing injuries compared to control and baseline. Compared to the experimental group, the odds ratio (95% CI) was **1.78 (1.04–3.04)** for experimental factory controls, and **1.77 (1.13–2.79)** for control factory controls
	Morgan et al., 2012 [31]	Individually randomized RCT	Overweight/obese (BMI 25–40) male aluminum workers in Australia Intervention: *n* = 65 Control: *n* = 45	Duration: October 2009 to March 2010 Extended follow up: 12 months	Didactic education, lifestyle feedback, free pedometers, and a monetary incentive to lose weight (although the stated goal was obesity reduction, injury prevention was an intended effect of intervention); controls were put on a waitlist	Significant effect in preventing injuries compared to control—the mean difference between groups was **0.3 fewer injuries per person for the intervention group (95% CI: 0.0–0.6)**
	Hermann et al., 2010 [36] ^1^	Controlled before and after	Automobile plant workers in Mexico Intervention: one plant (workforce 873–1350) Control: two plants (workforces 2990–3800 and 1291–1369 respectively)	Duration: January 1997 to January 2004 Extended follow up: N/A	Didactic education, a social marketing campaign, and behavioral feedback, plus administrative changes; controls received a basic safety campaign	Significant effect in preventing injuries compared to baseline—**92% percent decrease** of medical plus lost-time cases in the experimental plant, 3% in control plant A, and 6% in control plant B
	Mancini et al., 2005 [37]	Controlled before and after	Factory workers in Italy Intervention: *n* = 237 metal-ware factories (workforce not stated) Control: construction and wood/ceramics industries (workforce not stated)	Duration: December 1991 to June 1992 Extended follow up: 11 years	Didactic education and a social marketing campaign; controls received no intervention	Significant effect in preventing eye injuries compared to control—post-intervention incident rate ratios (95% CI) **were 0.77 (0.61–0.97) after 1–4 years, 0.63 (0.50–0.79) after 5–8 years, and 0.58 (0.43–0.77) after 9–11 years**
	Yeow & Goomas 2014 [38]	Controlled before and after study	Fluid plant workers in the USA Intervention: one fluid manufacturing plant, *n* = 362 Control: one fluid manufacturing plant, *n* = 338	Duration: 26 months Extended follow up: N/A	A safety-based monetary incentive program, peer-based monitoring and safety discussions; controls received didactic lectures only	Significant effect in preventing injuries compared to control **(48% reduction after 2 years)** and baseline **(33% reduction after 2 years)**
	Porru et al., 2011 [42]	Interrupted Time Series	Foundry workers in Italy One ferrous foundry (*n* = 230 approximately) and one non-ferrous foundry (*n* = 50 approximately)	Duration: 2000–2002 Extended follow up: 7 years	Safety discussions, didactic and participatory education, technical and organizational support, and health surveillance	Significant effect in preventing injuries in the short, medium, and long term for foundry A, but only in the long term for foundry B
	Porru et al., 2017 [43]	Interrupted time series	Foundry workers in Italy 22 ferrous (total *n* = 2750 workers) and 7 non-ferrous foundries (total *n* = 710 workers)	Duration: 2008 to 2013 Extended follow up: N/A	Improved formalization and dissemination of safety instructions, didactic education, safety audits and administrative support, and health surveillance	Only significant **26% (95% CI: 5–43%)** reduction in incidence rate (per worker) but not frequency rate (per hour) in ferrous foundries; no significant differences found in non-ferrous foundries
	Borger et al., 2011 (abstract) [40]	Interrupted time series	Glass factory workers in Germany 10 glass factories, *n* = 860	Duration: 2002 to 2003 Extended follow up: 6 years	Training on job-specific safety behaviors and risk management	Significant effect in preventing injuries—ITS reveals a **37% decrease** that can be attributed to intervention
	Shouman et al., 2002 [54]	Uncontrolled before and after	Glass factory workers in Egypt, *n* = 1229	Duration: 2000 calendar year Extended follow up: N/A	Didactic education, social marketing, a safety-based monetary incentive, greater availability of PPE, and administrative support	Significant effect in preventing injuries—**24% reduction** in both incidence rate (per worker) and frequency rate (per hour)
	Nunu et al., 2018 [55]	Uncontrolled before and after	Cement manufacturing workers in Zimbabwe, *n* = 244	Duration: 2007 to 2011 Extended follow up: N/A	Peer-based behavioral monitoring and reinforcement; rewards for safe behavior and reorientation for unsafe behavior	Significant effect in preventing injuries—**37% reduction** in the number of injuries
	Day 2002 (thesis) [56]	Uncontrolled before and after	Workers at a pulp and paper mill in Canada, *n* = 190	Duration: February to March 2002 Extended follow up: 2 months	One-day safety leadership course, all workers were welcome to attend	Although medical and first-aid incidents increased, statistical significance was not stated
	Gatti et al., 2013 (abstract) [48]	Uncontrolled before and after	Workers in 2 factories in Italy, sample sizes not stated	Duration: both studies are 2010 to 2012 Extended follow up: N/A	Behavioral feedback, reinforcement, and problem solving	Although injury rates **decreased by 52%** in the first factory and **68%** in the second factory, statistical significance was not stated in either
Mining, Quarrying, and Oil and Gas Extraction	Adams et al., 2013 [32]	Cluster-randomized RCT at the quarry level	Stone quarry workers in India Intervention: *n* = 103 from three experimental quarries Control: *n* = 101 from three control quarries	Duration: September 2006 to March 2007 Extended follow up: N/A	Eleven sessions of didactic education, social marketing, group motivational sessions, and individual counselling; controls received one session of standard didactic education	Significant effect in preventing injuries compared to baseline—**12% reduction (95% CI: 3–21%)**
	Hagge et al., 2016 [34]	Controlled before and after	Miners in the USA Intervention: *n* = approximately 400 Control: industry standard	Duration: 2007 to 2014 Extended follow up: N/A	Peer safety observation and feedback, and creation of a new safety plan, plus safety-oriented administrative changes	Although injury rates **decreased by 50%,** statistical significance was not stated
	Kowalski-Trakofler & Barrett 2016 [60]	Uncontrolled before and after	Miners in the USA Study B: 4 mines with >2300 workers total; Study C: 1 mine with 550 workers (Study A not included due to lack of injury outcomes)	Duration: 1995 to 1996 for Study B; 1995 for Study C Extended follow up: 12 months for Study C	Degraded images were used instead of highlighted images during safety training	Significant effect in preventing injuries in study B (**9.06% decrease in the first year and a further 29.94% in the second year**); although injury rates **decreased by 27.1%** in study C, significance was not stated
Public Administration	Sullivan et al., 2017 [28]	Matched-pairs cluster RCT, randomized at the station level	Firefighters in the USA Intervention: 16 stations, *n* = 601 Control: 16 stations, *n* = 588	Duration: last 2 weeks of August 2009 Extended follow up: 54 weeks	Didactic education on sleep health (although the stated goal was to improve sleep health, injury prevention was an intended effect of intervention); controls did not receive intervention	Significant effect in reducing the number of injury and disability days (**1.4 per worker in the intervention group vs. 2.6 per worker in the control group**), but not the number of injuries and motor vehicle crashes
	Knapik et al., 2004 [44]	Historically controlled	Soldiers in the USA Intervention: *n* = 1283 (1122 men and 161 women) Historical control: *n* = 2559 (2303 men and 256 women)	Duration: 36 weeks Extended follow up: N/A	Modified physical training and didactic education, plus administrative injury surveillance support	Significant effect in preventing injuries for men only—adjusted risk ratio of control vs. intervention (95% CI) was **1.50 (1.06–2.12)** for men and **1.37 (0.57–3.29)** for women
Transportation and Warehousing	Robaina et al., 2010 [35]	Controlled before and after	Stevedores in Cuba Intervention: *n* = 185 (from one terminal) Control: *n* = 105 (from another terminal)	Duration: January 2004 to April 2005 Extended follow up: until end of 2006	Group safety discussions, didactic and participatory education for workers and supervisors; controls received no intervention.	Significant effect in preventing injuries compared to control (**58.8% of injuries prevented**) and baseline (**2.8 fewer injuries** per 100 person-years)
	Howard et al., 2009 (abstract) [57]	Uncontrolled before and after	Road transport drivers, Australia, *n* = 800	Duration: 3 years Extended follow up: 12 months	Sleep health education and individual health screening, (although the stated goal was to improve sleep health, injury prevention was an intended effect of intervention)	Significant effect in preventing injuries—lost time injuries per 100 drivers were **reduced from 17.1 to 14.2**
Utilities	Salminen 2008 [49]	Uncontrolled before and after	Electricians in Finland, 1998 to 2005 Study 1: *n* = 172 Study 2: *n* = 179	Duration: 2001 to 2002 for Study 1; 2001 for Study 2 Extended follow up: 3 years for both studies	Study 1: group safety discussion followed by collaborative decision on solutions Study 2: Didactic and participatory education on driving safety	In Study 1, although work-related traffic incidents **decreased by 72.2%**, other occupational injuries **increased by 15.1%.** The proportion of traffic-related incidents decreased significantly. In Study 2, although the rate of injuries ***increased* by 50%,** statistical significance was not stated
	Badmos 2018 (abstract) [58]	Uncontrolled before and after	Electricity distribution company employees in Nigeria, (sample size not stated)	Duration: 2015 to 2017 Extended follow up: N/A	Safety counselling and videos, safety huddles, and hazard identification competitions, plus administrative changes	Although injury rates **decreased by 40%** among staff, statistical significance was not stated.
Mixed Industries	Boini et al., 2017 [45]	Prospective cohort	Young workers in France, 2009–2014 Exposed: students who received occupational safety and health (OSH) training in school, *n* = 687 Unexposed: students who did not receive training, *n* = 68	Duration: variable Extended follow up: 2 years	Didactic education (varied based on type of diploma)	Significant effect in preventing injuries—the incident rate ratio of exposed to unexposed was **0.51 (95% CI: 0.00–0.98).**

^1^ Not included in narrative synthesis due to critical risk of bias.

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
