# Peer review of "In Search of Concrete Outcomes—A Systematic Review on the Effectiveness of Educational Interventions on Reducing Acute Occupational Injuries"

_ijerph, 2020, doi:10.3390/ijerph17186874_

Round 1

Reviewer 1 Report

Dear authors,

thank you for the opportunity to assess this systematic review.

The research question is very relevant and methods as well as presentation of results is very sound in general.

I would suggest some minor ammendments to this review:

line 60: please add a reference to PRISMA statement

line 107: you have excluded studies before 2000 but you explanation does not quite match to you research questions. Was there a dramatic change in the way workers were adviced in occupational safety?

line 155: you state that meta-analysis was not attemted due to heterogeneity which is legit in general. However meta-analysis could have been done with studies of same type and same industry. As far as I know differences in risk of bias do not prevent authors of Cochrane to do meta-analysis but adding sensitivity analysis instead. However you should state if meta-analysis was not possible due to insufficent number of studies from same industries, to show that at least you tried to do so.

line 167: you have excluded some studies as they are duplicates of included studies. I suggest to add these studies to included study section e.g. with footnotes instead of excluding them.

Good luck

Author Response

Thank you for your thoughtful suggestions.

Point 1: Dear authors,

thank you for the opportunity to assess this systematic review.

The research question is very relevant and methods as well as presentation of results is very sound in general.

I would suggest some minor ammendments to this review:

line 60: please add a reference to PRISMA statement

Response 1: Thank you for taking the time to review our paper. We greatly appreciate your helpful feedback and suggestions. We have included a reference to the PRISMA statement on line 63.

Point 2: line 107: you have excluded studies before 2000 but you explanation does not quite match to you research questions. Was there a dramatic change in the way workers were adviced in occupational safety?

Response 2: Thank you for bringing this up. We have added a sentence on lines 113-114 to confirm that guidelines for occupational safety education was changing around the year 2000, and the sources were cited accordingly.

Point 3: line 155: you state that meta-analysis was not attemted due to heterogeneity which is legit in general. However meta-analysis could have been done with studies of same type and same industry. As far as I know differences in risk of bias do not prevent authors of Cochrane to do meta-analysis but adding sensitivity analysis instead. However you should state if meta-analysis was not possible due to insufficent number of studies from same industries, to show that at least you tried to do so.

Response 3: Thank you for this suggestion. We have modified section 2.10 (specifically lines 159-160) to clarify that an insufficient number of studies from most industries precluded meta-analysis.

Point 4: line 167: you have excluded some studies as they are duplicates of included studies. I suggest to add these studies to included study section e.g. with footnotes instead of excluding them.

Response 4: Thank you for catching this. We have edited section 3.1 accordingly (specifically lines 172-175 and Figure 1).

Reviewer 2 Report

In my opinion, the manuscript is very interesting and important because accidents at work are a big social and economic problem and the selection of appropriate methods of reducing them is a challenge for lawmakers, employers and OHS services. However, in general, as the authors wrote in the Inroduction, the effectiveness of injury prevention is often assessed based on subjective measures, instead of direct injury outcomes analysis. The authors decided to limit their Review to publications based on the true effectiveness of education on reducing workplace injuries, because this problem remains unclear.

The review was carried out very carefully, using the current recommendations for both the selection of literature and the evaluation of possible bias sources. The discussion of the results is comprehensive and allows to define the needs and directions of further research as well as to indicate the branches of the economy and countries where preventive measures are not taken or the access to knowledge on this subject is limited.

I have only some minor remarks. I suggest to change a title, because Authors don’t analyse the possiblities of education the workers, but the possibilities to check results of this education. Therefore first part of title does not fit to the content of the paper. In lines 235/236 Authors wrote „This  limits our ability to generalize the results to the rest of the industry”. In my opinion arts, enertainment and recreation are rather „sector of economy” than industry. The same in Line 262 (health care and social assistance).

Author Response

Thank you for your thoughtful comments.

Point 1: In my opinion, the manuscript is very interesting and important because accidents at work are a big social and economic problem and the selection of appropriate methods of reducing them is a challenge for lawmakers, employers and OHS services. However, in general, as the authors wrote in the Inroduction, the effectiveness of injury prevention is often assessed based on subjective measures, instead of direct injury outcomes analysis. The authors decided to limit their Review to publications based on the true effectiveness of education on reducing workplace injuries, because this problem remains unclear.

The review was carried out very carefully, using the current recommendations for both the selection of literature and the evaluation of possible bias sources. The discussion of the results is comprehensive and allows to define the needs and directions of further research as well as to indicate the branches of the economy and countries where preventive measures are not taken or the access to knowledge on this subject is limited.

I have only some minor remarks. I suggest to change a title, because Authors don’t analyse the possiblities of education the workers, but the possibilities to check results of this education. Therefore first part of title does not fit to the content of the paper.

Response 1: Thank you for taking the time to review our paper. We greatly appreciate your helpful feedback and suggestions. We have changed the title to “In search of concrete outcomes – a systematic review on the effectiveness of educational interventions on reducing acute occupational injuries”. We believe this better reflects the outcome-oriented nature of our review.

Point 2: In lines 235/236 Authors wrote „This  limits our ability to generalize the results to the rest of the industry”. In my opinion arts, enertainment and recreation are rather „sector of economy” than industry. The same in Line 262 (health care and social assistance).

Response 2: Thank you for this suggestion. We have changed the wording accordingly (see lines 251 and 277 respectively).

Reviewer 3 Report

This is a well-written review. I only have one comment. If possible, I recommend a table or a figure to demonstrate the effect size of education on occupational injury prevention. Of course, a quantitative meta-analysis would be more informatic to help the readers understand the results.

Author Response

Thank you for reviewing our paper.

Point 1: This is a well-written review. I only have one comment. If possible, I recommend a table or a figure to demonstrate the effect size of education on occupational injury prevention. Of course, a quantitative meta-analysis would be more informatic to help the readers understand the results.

Response 1: Thank you for taking the time to review our paper. The effect measures reported by the included studies are highly heterogenous. For example, some studies report odds ratios while others report rate ratios, some studies measure injuries per worker while others measure injuries per man-hour, and some studies compare the experimental group to baseline while others compare it to control group(s), etc. These areas of heterogeneity pose a significant challenge for us to accurately summarize effect sizes using a figure. Nevertheless, we have highlighted the effect sizes in Table 2 using bold font to make it easier for readers to appreciate the effectiveness of educational interventions at a glance.

Of course, in future publications we will aim to perform meta-analyses whenever possible.

Reviewer 4 Report

The authors made a very detailed review of the literature in the field of workplace injuries prevention. The methodology, including the justification for the actions taken to implement this systematic review, was described in detail.

This review could help in planning and selecting the appropriate form of education for employees in a given industry and increase  effectiveness of such interventions.

Author Response

Dear Reviewer,

Thank you for taking the time to review our paper. We appreciate your encouragement and validation. Hopefully the findings described in this paper will help guide future injury prevention programs and enhance worker safety.